# Reciprocal modulation of ammonia and melanin production has implications for cryptococcal virulence

Rosanna P. Baker [1] & Arturo Casadevall [1] ✉

The fungus *Cryptococcus neoformans* is the causative agent of cryptococcosis, a disease that is uniformly lethal unless treated with antifungal drugs, yet current regimens are hindered by host toxicity and pathogen resistance. An attractive alternative approach to combat this deadly disease is the direct targeting of pathogen-derived virulence mechanisms. *C. neoformans* expresses multiple virulence factors that have been studied previously as isolated entities. Among these, are urease, which increases phagosomal pH and promotes brain invasion, and melanization, which protects against immune cells and antifungal treatments. Here we report a reciprocal interdependency between these two virulence factors. Cells hydrolyzing urea release ammonia gas which acts at a distance to raise pH and increase melanization rates for nearby cells, which in turn reduces secretion of urease-carrying extracellular vesicles. This reciprocal relationship manifests as an emergent property that may explain why targeting isolated virulence mechanisms for drug development has been difficult and argues for a more holistic approach that considers the virulence composite.

Virulence factors are traits that confer upon infectious microbes the capacity to damage and persist in infected hosts despite immune responses[1]. Although pathogens tend to employ multiple virulence factors, including surface coatings, toxins, and enzymes, most studies focus on the independent contribution to pathogenicity of a single factor without consideration of its contribution to the virulence composite[2]. The mechanisms by which virulence factors interact is a relatively unexplored topic in microbial pathogenesis.

*C. neoformans* expresses a diverse suite of virulence factors that include a polysaccharide capsule, melanin production and the expression of various enzymes such as urease and phospholipase, among others. The polysaccharide capsule and melanin pigment protect against phagocytosis and phagocytic cell oxidative bursts, respectively[3,4]. Urease is a nutritional enzyme, which incidentally interferes with phagosomal acidification by hydrolyzing urea to generate ammonia[5] and plays a crucial role in brain invasion[6,7], while phospholipase damages macrophage phagosomal membranes and

promotes intracellular survival[8]. There is little information on how these virulence factors interact with one another.

This study characterizes the interaction between cryptococcal urease activity and melanization. Urease is released in extracellular vesicles and hydrolyzes urea to produce ammonia, which is shown here to mediate action at a distance through its ability to travel as a gas. The melanization reaction is dependent on pH and thereby promoted by increased production of ammonia by urease. Enhanced melanization was found to favor persistence of cryptococcal cells within phagosomes and thereby increase brain dissemination through a Trojan horse mechanism. Melanin deposition in the cell wall acts as a feedback mechanism by reducing extracellular vesicle production to inhibit the release of vesicle-associated urease. Hence, urease and melanin exhibit reciprocal modulation that provide advantages at different stages of infection and this coordination and synergy of virulence factors results in phenotypic changes that could not have been anticipated from studying each component in isolation.

[1]Department of Molecular Microbiology and Immunology, Johns Hopkins Bloomberg School of Public Health, Johns Hopkins University, Baltimore, MD 21205, USA. ✉e-mail: acasade1@jhu.edu

## Results

### *C. neoformans* urease produces ammonia that acts at a distance

While assaying urease activity of *C. neoformans* in rapid urea broth[9], we noticed that with extended incubation time, media in cell-free wells also increased in pH (Fig. 1a, left panel). Measurement of absorbance at 560 nm ($A_{560}$) for each well revealed a direct linear relationship between the color change in the cell-free well and the number of cells in the adjacent well (Fig. 1a, right panel). To determine if this phenomenon was dependent on urease activity, either wild-type (WT) or urease-deficient (*ure1Δ*) cells were grown in urea broth in one well of a 24-well plate along with cell-free media in the remaining 23 wells. After incubation at 30 °C for 24 h, media in the well containing WT cells and several surrounding wells changed color from yellow to pink while no color change was observed for the plate containing *ure1Δ* cells (Fig. 1b, left panel). The sigmoidal relationship between $A_{560}$ of cell-free media and the distance to WT cells producing ammonia (Fig. 1b, right panel)

resembled a pH titration curve for which the midpoint is the pKa. A Boltzmann curve fit of the data gives a midpoint value of 56 mm and suggests that the pH of media in eight wells within this distance has increased to at least 7.9, the pKa of phenol red in urea broth. Using a hand-held ammonia gas detector with a range of 0–200 ppm, ammonia was detectable during 30 s readings for cultures grown in urea broth for 24 h at 30 °C with a minimal starting cell density of $2 \times 10^6$ cells/mL when the urea concentration was held constant at 2% (Fig. 1c, left panel) or with a minimal urea concentration of 0.25% and constant cell density of $1 \times 10^8$ cells/mL (Fig. 1c, right panel). In both cases, the cumulative amount of ammonia produced during 24 h was considerably higher as evidenced by the increased pH of the culture media at even lower cell densities and urea concentrations (Fig. 1c).

### Released ammonia promotes melanization and decreases growth rate of nearby cells

We next explored how ammonia released from *C. neoformans* cells would impact nearby cells. Since chemical oxidation of dopamine occurs more readily at alkaline pH[10], we posited that increased pH due to ammonia production might promote melanin formation. Indeed, cells growing on dopamine-supplemented agar near WT compared to *ure1Δ* cells in urea broth showed greater melanin pigment production that increased linearly with proximity to the source of ammonia (Fig. 2a). As *C. neoformans* grows best at acidic pH[11], melanization is routinely assayed in pH 5.5 minimal media. When the pH of dopamine-supplemented agar was titrated from pH 5.5 to 7, pigment production increased significantly from pH 6 to 6.5 and from pH 6.5 to 7 (Fig. 2b). This suggests that the increased melanization observed for cells growing near ammonia-producing cells was the result of increased pH of the dopamine-agar. Whereas ammonia production by WT cells growing in urea broth stimulated melanization of nearby cells growing on pH 5.5 dopamine-agar, pre-adjustment of the media to pH 7 yielded comparable levels of pigmentation for WT, *ure1Δ* and cell-free control plates (Fig. 2c, left panel). After 48 h at 30 °C, the pH of the dopamine-agar increased significantly only in the plate containing WT cells in urea broth, but not for plates containing *ure1Δ* or cell-free controls (Fig. 2c, right panel). We also examined the effect of ammonia release on the growth of *C. neoformans* in nearby wells and found that media pH increased proportionally with increasing proximity to the ammonia source and when pH increased to 8.5 or higher, growth was impaired (Fig. 2d).

### Laccase 1 expression and localization are impacted by ammonia

To ascertain whether the increased pigmentation produced in response to ammonia was indeed attributable to melanin production by *C. neoformans* or merely increased auto-polymerization of dopamine which is also expedited by alkaline pH[10], we extended our analysis to include mutant strains defective in melanin production. In response to glucose starvation, *C. neoformans* expresses two laccase genes, *LAC1*, the primary laccase gene, deletion of which results in an albino phenotype[12], and the 75% identical homolog, *LAC2*, that is expressed at a lower level and has only a minor melanization defect when deleted[13]. We assayed WT, *lac1Δ* and *lac2Δ* cells for melanization on DA-supplemented agar in the presence of WT or *ure1Δ* cells in urea broth and found that ammonia-mediated stimulation of pigment production was abrogated for *lac1Δ* but not for *lac2Δ* (Fig. 2e). Thus, laccase 1, the primary laccase responsible for melanization in *C. neoformans* is also responsible for the melanization effects related to diffusible ammonia.

Since melanization catalyzed by laccase 1 was specifically stimulated by urease-produced ammonia, we proceeded to investigate the impact of ammonia gas on laccase 1 expression and cellular localization. Expression of an amino-terminal GFP-laccase 1 fusion protein revealed that the enzyme is mis-localized in cytoplasmic vesicles at acidic pH and trafficked to the cell wall under physiological pH[14]. We grew the GFP-laccase 1 expression strain in all wells of a 24-well plate except for the

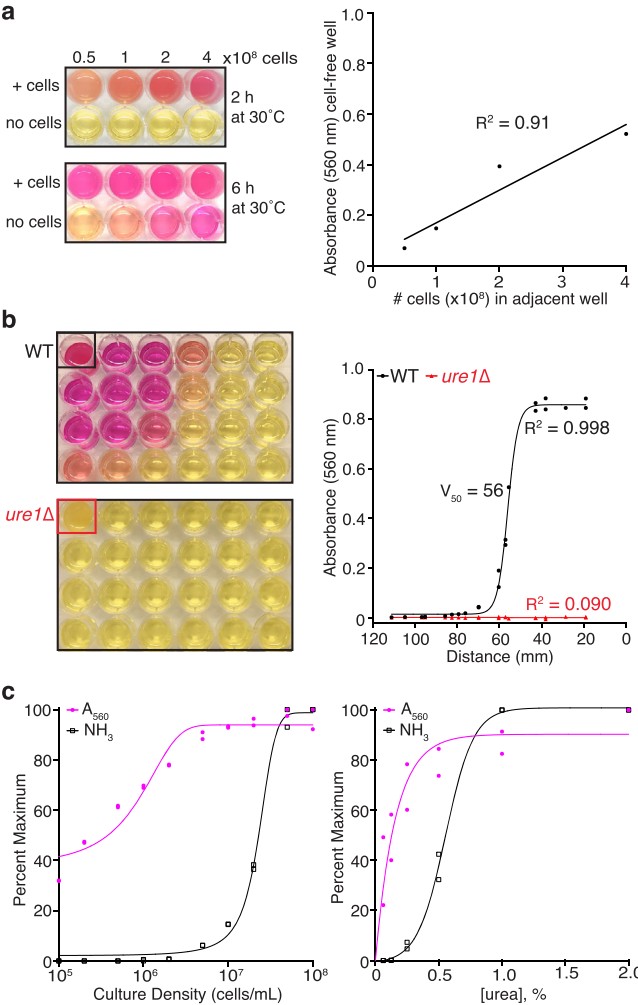

**Fig. 1 | *C. neoformans* urease produces ammonia that acts at a distance. a** Color photographs (left panel) of wells containing urea broth with the indicated number of *C. neoformans* cells or no cells following incubation at 30 °C for 2 or 6 h. Absorbance at 560 nm ($A_{560}$) of cell-free media after 6 h shows a linear increase with increasing numbers of cells in adjacent wells (right panel). **b** Plates containing urea broth in all wells and $1 \times 10^8$ wild-type (WT) or *ure1Δ* cells in the top left well (A1) photographed after 24 h at 30 °C (left panel). A sigmoidal relationship between $A_{560}$ and proximity to well A1 (right panel) is observed for WT but not urease-deficient cells. **c** Percent maximum $A_{560}$ of culture supernatant (pink) and ammonia gas ($NH_3$, black) measured for WT cultures grown in urea broth at 30 °C for 24 h with increasing starting cell density (left panel) or increasing urea concentration (right panel). Source data are provided as a Source Data file.

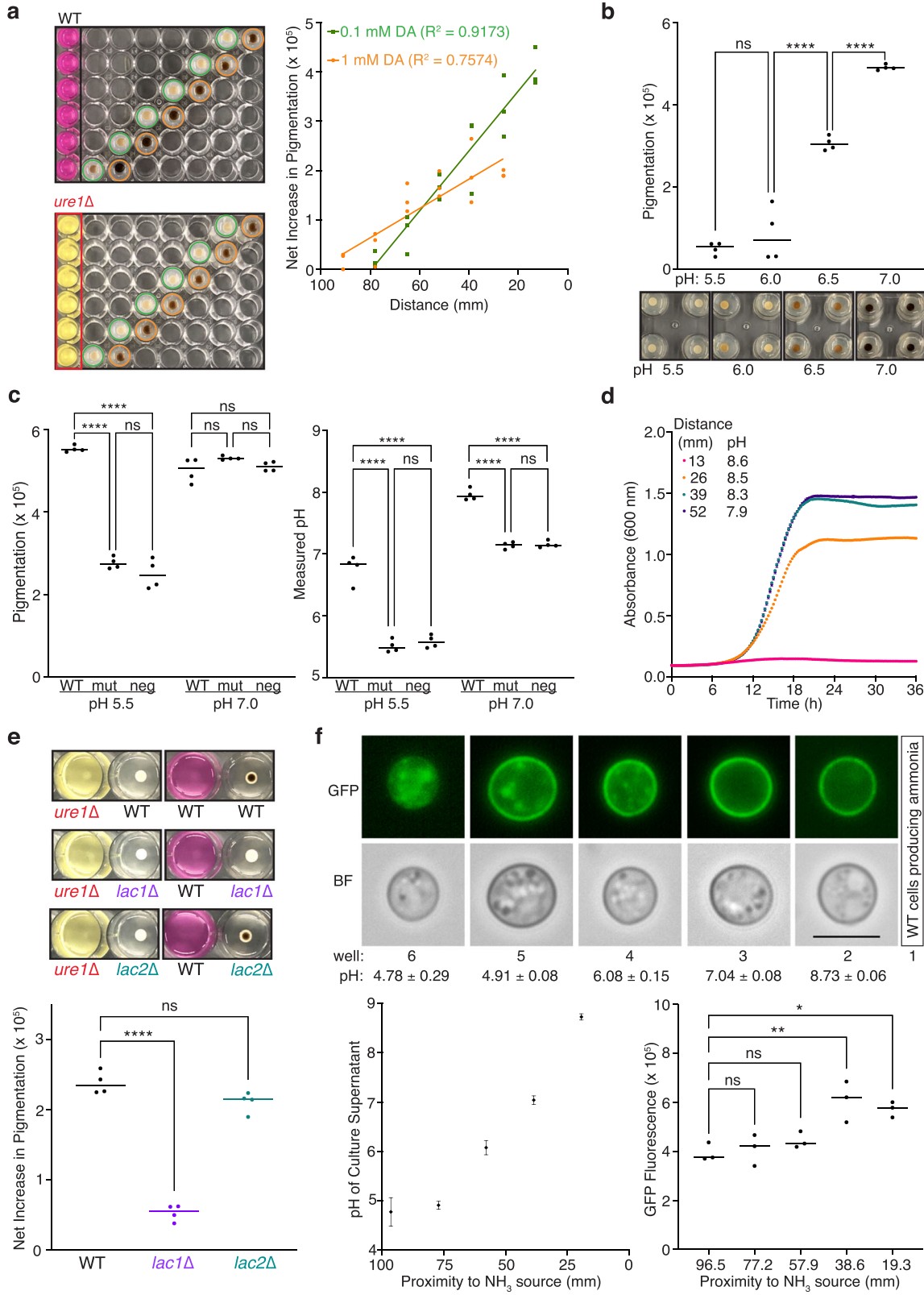

first vertical column of wells, to which WT cells in urea broth were added. Following incubation at 30 °C for 24 h, cellular localization of GFP-laccase 1 was visualized by fluorescence microscopy of live cells. With increasing proximity to the source of diffusible ammonia, GFP-laccase 1 became increasingly targeted to the cell wall (Fig. 2f, upper panels) and this correlated with increasing pH of the culture media (Fig. 2f, lower left panel). Spectrophotometric measurement of GFP fluorescence revealed that cells in the wells closest to the ammonia

source had significantly more fluorescence than more distant cells (Fig. 2f, lower right panel), which is consistent with greater GFP-laccase 1 expression in response to increased alkalinity of the culture media.

**Melanization decreases urease-mediated ammonia production by *C. neoformans***

Whereas ammonia produced by urease activity promoted melanization, melanization in turn dramatically decreased urease activity

**Fig. 2 | Ammonia release increases local pH to stimulate melanization and decrease growth rate. a** WT *C. neoformans* cells melanized for 24 h at 30 °C on agar supplemented with 0.1 (green) or 1 mM (orange) dopamine (DA) in the presence of WT or *ure1Δ* cells in urea broth (left panels). Quantified pigmentation for three biological replicates reveals a linear net increase in melanization with increasing proximity to ammonia-producing WT cells compared to the urease-deficient mutant (right panel). **b** WT cells melanized for 48 h at 30 °C on 1 mM DA-supplemented agar show significantly increased pigmentation at pH 6.5 and 7.0. *n* = 4, NS not significant, ****$p < 0.0001$; one-way ANOVA with Tukey's multiple comparisons test. **c** Quantified pigmentation (left panel) and pH of agar (right panel) following growth of WT *C. neoformans* at 30 °C for 48 h on pH 5.5 or 7.0 agar supplemented with 1 mM DA adjacent to a well containing WT or *ure1Δ* (mut) cells in urea broth or cell-free media (neg). *n* = 4, NS not significant, ****$p < 0.0001$; one-way ANOVA with Tukey's multiple comparisons test. **d** Growth curves of cells growing at the indicated distance from cells in urea broth. Graphed is the mean ± standard deviation of three biological replicates. The pH of culture supernatants measured at the end of the experiment is indicated. **e** Representative images (upper panels) of WT, *lac1Δ*, or *lac2Δ* cells grown on 1 mM DA-supplemented agar near WT or *ure1Δ* cells in urea broth. Graphed is the net increase in pigmentation calculated by subtracting measurements for each well in a *ure1Δ*-plate from that of the corresponding well in a WT-plate (lower panel). *n* = 4, NS not significant, ****$p < 0.0001$; one-way ANOVA with Tukey's multiple comparisons test. **f** Fluorescence (GFP) and bright field (BF) images (upper panels) of cells expressing GFP-laccase 1 after incubation at 30 °C for 24 h in columns 2 through 6 of a 24-well plate containing WT cells in urea broth in column 1 (scale bar = 5 μm). The pH of culture supernatants increased with increasing proximity to cells producing ammonia (lower left panel, graphed as mean ± standard deviation of 3 biological replicates) and GFP fluorescence levels were significantly higher in the two columns nearest the ammonia source (lower right panel). *n* = 3, NS not significant, *$p = 0.0154$, **$p = 0.0046$; one-way ANOVA with Tukey's multiple comparisons test. Source data are provided as a Source Data file.

(Fig. 3a). Melanization impedes the uptake of liposomes into cells[15] and we found that the melanized cell wall similarly imparts a barrier to the release of extracellular vesicles as evidenced from the lower fluorescence of a lipophilic probe in culture supernatants of melanized compared to non-melanized cells (Fig. 3b). To explore the impact of melanization on vesicular transport more directly, we employed a recently optimized method for the isolation extracellular vesicles from cells grown on solid media[16] to compare the yields from non-melanized and melanized cells. Both lipid content and urease activity were significantly lower for vesicle preparations from comparable numbers of WT cells grown on agar supplemented with 1 mM dopamine compared to minimal media alone (Fig. 3c). For the *lac1Δ* strain that fails to produce melanin pigment, extracellular vesicle yield and urease activity were comparable for cells grown in the absence or presence of dopamine (Fig. 3c). Thus, melanin in the cryptococcal cell wall impedes vesicular transport from cells and this results in lower urease activity because urease is one of the virulence factors secreted in extracellular vesicles[17]. Diminished urease activity of melanized compared to non-melanized cells also manifested in a significant decrease in the amount of ammonia released (Fig. 3d) and the ability of melanized cells to stimulate melanization of nearby cells (Fig. 3e).

## Urease-positive *C. neoformans* produce ammonia that promotes melanization in lung tissue

To investigate the impact of urease-produced ammonia during infection, mice were inoculated with approximately $5 \times 10^5$ WT or *ure1Δ* cells by intranasal inhalation and the infection was allowed to proceed for 3 weeks. At this time, the animals infected with WT were noticeably moribund compared to *ure1Δ*-infected mice, which appeared healthy. Comparing fungal burden in the lungs revealed an average of $6.1 \times 10^8$ colony forming units (CFU) for WT, which was more than an order of magnitude greater than the average of $3.1 \times 10^7$ measured for *ure1Δ* (Fig. 4a). For two of the ten mice infected with *ure1Δ* cells, a productive lung infection did not develop so these animals were excluded from further analysis. The lungs dissected from WT-infected mice were significantly more massive compared to the lungs recovered from *ure1Δ*-infected mice (Fig. 4b). Consistent with the role of urease activity in brain dissemination[6,7], the brain fungal burden in mice infected intranasally with *ure1Δ* was approximately four logs lower than in the lungs, whereas CFU for WT-infected mice were on average only about 100-fold lower in brains compared to lungs (Fig. 4a).

We postulated that urease-positive *C. neoformans* would produce detectable levels of ammonia during acute pulmonary infection. By enclosing infected animals in a small airtight container for 2 min, an average reading of $1.1 \pm 0.5$ ppm ammonia gas was measured in the breath of WT-infected animals, compared to *ure1Δ*-infected mice for which ammonia gas was undetectable (Fig. 4c). By way of comparison, a comparable ammonia reading of ~1.5 ppm was measured for $2 \times 10^6$ *C.*

*neoformans* cells growing in the presence of 2% urea (Fig. 1c). We extended our investigation by measuring the pH of infected lungs after dissection by probing the tissue with a needle pH electrode. These measurements revealed significantly higher pH in the lungs of WT compared to *ure1Δ* mice (Fig. 4d), which suggests that urease-positive animals produce sufficient ammonia to increase the pH of infected tissue. To control for possible regional variations in pH throughout the organ, dissected lung tissue was homogenized in sterile saline solution and the pH of the resulting cell-free supernatant was measured using a standard pH electrode. Consistent with the in-situ measurements, the pH of lung tissue homogenates was significantly higher for those derived from WT compared to *ure1Δ* infected animals (Fig. 4d). Our results reveal a previously unanticipated consequence of *C. neoformans* infection, namely, that urease activity releases gaseous ammonia that alters the cellular environment of infected tissue by increasing pH.

Since melanization was increased at higher pH values in vitro (Fig. 2), we hypothesized that urease-dependent increase in lung pH would similarly promote melanization in vivo. Unfortunately, silver-based stains do not reliably distinguish melanized from non-melanized *C. neoformans* cells because the stains also react with polysaccharides in the cell wall[18,19]. Despite this limitation, it was possible to discern deeply pigmented *C. neoformans* cells from the surrounding tissue in hematoxylin and eosin (H & E) stained sections of infected *Galleria mellonella*[20]. Similarly, we were able to detect a small number of heavily melanized cells in H & E stained sections of lung tissue from mice infected with WT or *ure1Δ* (Fig. 4e, upper panel). As layers of melanin are progressively deposited in the cell wall over time[21], the number of heavily pigmented cells detected is likely a representative underestimate of the total number of melanized cells. A significantly greater number of melanized cells were identified in stained sections of lung tissue that had been infected with WT compared to *ure1Δ* cells (Fig. 4e, lower left panel). To account for the 10-fold difference in lung CFU for WT- and *ure1Δ*-infected animals, equivalently sized areas surrounding each melanized cell were quantified for the total number of cryptococcal cells. This analysis revealed a significantly higher percentage of melanized cells for WT compared to *ure1Δ* (Fig. 4e, lower right panel). Cryptococcal melanin is acid stable[22] and we detected characteristic acid-resistant melanin 'ghost' particles after boiling lung tissue samples for 1 h in concentrated hydrochloric acid (Fig. 4f) that were approximately 3-fold more abundant for WT compared to *ure1Δ*. The increased number of melanized WT compared to *ure1Δ* cells in infected lung tissue is consistent with our in vitro observations that melanization was promoted by urease-dependent ammonia production.

## Melanization promotes brain dissemination of *C. neoformans*

As melanization increases survival of *C. neoformans* during coincubation with macrophages[4], we sought to extend this analysis by infecting

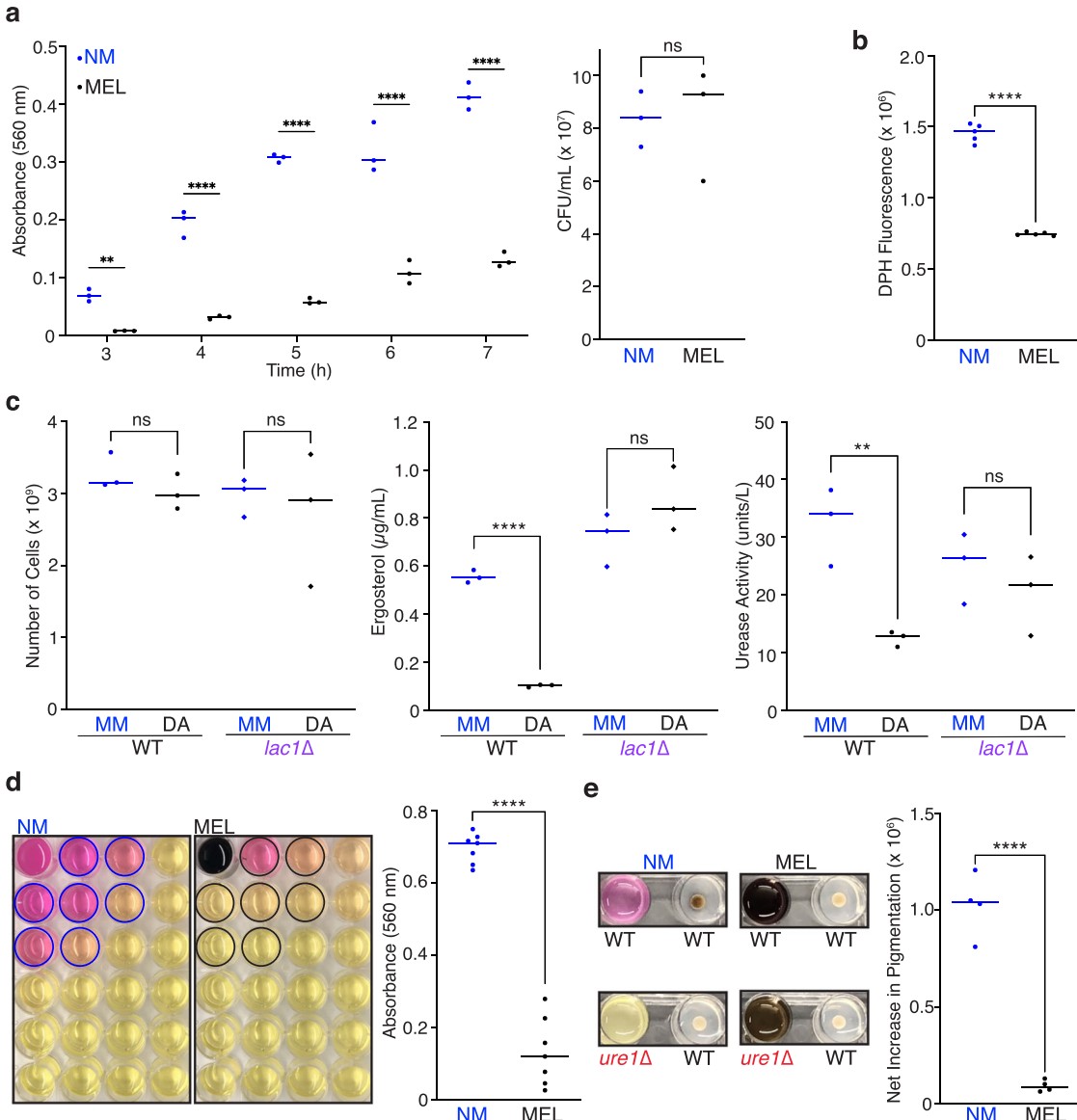

**Fig. 3 | Melanized cells release less ammonia and fail to promote melanization of neighboring cells. a** Urease activity measured as $A_{560}$ of urease broth culture supernatants over time was significantly higher for non-melanized (NM) compared to melanized (MEL) cells (left panel; $n = 3$, **$p = 0.0045$, ****$p < 0.0001$; two-way ANOVA with Sidak's multiple comparisons test) for triplicate cultures with statistically comparable cell densities (right panel; $n = 3$, NS not significant; unpaired, two-sided parametric $t$-test). **b** Decreased release of extracellular vesicles from MEL compared to NM cells as measured by fluorescence of the lipophilic probe, 1, 6-diphenyl-1, 3, 5-hexatriene (DPH), in culture supernatants. $n = 5$, ****$p < 0.0001$; unpaired, two-sided parametric $t$-test. **c** Isolation of extracellular vesicles from comparable numbers of cells (left panel; $n = 3$, NS not significant) grown on minimal media (MM) or dopamine-supplemented (DA) agar plates yielded significantly lower lipid content (middle panel; $n = 3$, NS not significant, ****$p < 0.0001$; unpaired,

two-sided parametric $t$-test) and urease activity (right panel; $n = 3$, NS not significant, **$p = 0.0074$; unpaired, two-sided $p$arametric $t$-test) for WT but not $lac1\Delta$ cells grown in the presence of DA. **d** 24-well plates (left) with NM or MEL cells in urea broth in well A1 (upper left) and pH-sensitive media in remaining wells photographed after incubation at 30 °C for 24 h. Scatter plot (right) of $A_{560}$ for wells within 50 mm (outlined with circles) of well A1 indicates significantly reduced ammonia production for melanized cells. $n = 7$, ****$p < 0.0001$; unpaired, two-sided $p$arametric $t$-test. **e** WT cells grown for 18 h at 30 °C on 1 mM DA-supplemented agar in wells adjacent to NM or MEL WT or $ure1\Delta$ cells in urea broth (left panels). Graph (right) of the quantified net increase in pigmentation reveals significantly greater stimulation of melanization by NM compared to MEL WT cells. $n = 4$, ****$p < 0.0001$; unpaired, two-sided parametric $t$-test. Source data are provided as a Source Data file.

mice intravenously with macrophages harboring either non-melanized or melanized *C. neoformans* cells. We measured significantly higher fungal burdens in the lungs and brains of animals 24 h post-infection with melanized compared to non-melanized cells (Fig. 5a). This observation supports a model whereby melanized cells can persist longer in phagolysosomes than non-melanized cells. We also compared infections with macrophages carrying non-melanized and melanized *ure1Δ* cells and although CFU were comparable in lung tissue, there was a significantly larger number of melanized compared to non-melanized cells in the brain (Fig. 5b).

## Discussion

*C. neoformans* has evolved numerous protective properties that promote survival in habitats like soil and tree bark where these protect against ameboid predators[23]. Given similarities between amoeba and macrophages in their interaction with this fungus[24], many of these also function as virulence factors. For example, the polysaccharide capsule provides protection from desiccation in the environment and allows cryptococcal cells to evade attack by the host immune system[25]. Several other factors are secreted in extracellular vesicles dubbed 'virulence bags'[17], including urease, that hydrolyzes urea to produce

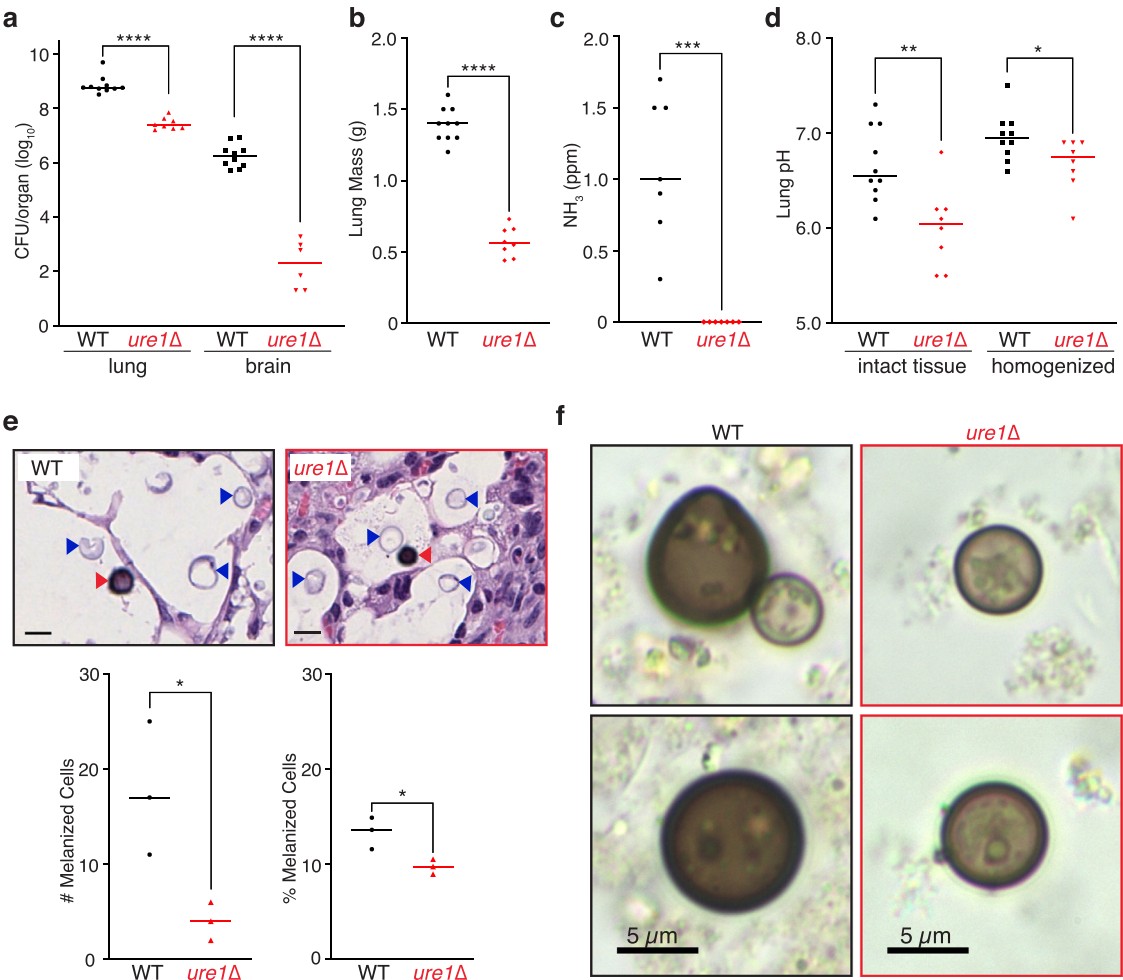

**Fig. 4 | Pulmonary murine infection model reveals urease-dependent ammonia production and increased pH and melanization in lung tissue. a** Intranasal infection resulted in a significantly higher fungal burden in both the lungs and brains of mice infected with WT ($n = 10$) compared to $ure1\Delta$ ($n = 8$, lung; $n = 6$, brain). ****$p < 0.0001$; unpaired, two-sided parametric $t$-test. **b** Lung mass was also significantly higher for mice infected intranasally with WT compared to $ure1\Delta$. $n = 10$, WT; $n = 8$, ure1Δ, ****$p < 0.0001$; unpaired, two-sided parametric $t$-test. **c** Ammonia levels measured in the exhalant of WT- or $ure1\Delta$-infected mice enclosed in a 55 cm³ enclosure for 2 min. $n = 7$, ***$p = 0.0001$; unpaired, two-sided parametric $t$-test. **d** Lung tissue pH measured using an MI-407 needle electrode immediately after dissection and after homogenization and centrifugation of tissue was

significantly higher for mice infected with WT compared to $ure1\Delta$ ($n = 10$, WT; $n = 8$, $ure1\Delta$, **$p = 0.0036$ and *$p = 0.0358$, respectively). **e** Representative images (upper panels) of H & E stained lung tissue sections from three mice infected with either WT or $ure1\Delta$ (blue arrows; scale bars = 10 μm). Quantification of the total number (lower left panel; $n = 3$, *$p = 0.0316$; unpaired, two-sided parametric $t$-test) and percentage (lower right panel; $n = 3$, *$p = 0.0274$; unpaired, two-sided parametric $t$-test) of heavily melanized cryptococcal cells (red arrows) detected in tissue sections from three infected animals. **f** Representative bright field images from three independent preparations of melanin-rich particles isolated from the lungs of mice infected with either WT or $ure1\Delta$ cells after boiling homogenized tissue in concentrated hydrochloric acid. Source data are provided as a Source Data file.

ammonia[26], and laccase, that catalyzes the oxidation of phenolic precursors to produce melanin[27]. Efforts to treat or prevent cryptococcosis have largely focused on targeting one of these factors, for instance, vaccines based on the glucuronoxylomannan (GXM) sugar moiety motifs of the capsule[28] or antibodies raised against melanin[29]. Despite concerted efforts on these fronts[30], an effective vaccine has not yet been developed and current antifungal drug regimens are not fully protective[31]. Given the multifaceted nature of *C. neoformans* virulence, development of successful disease treatment or prevention strategies may benefit from a combinatorial approach that considers the possible interaction between individual virulence factors.

Urease is a widely expressed enzyme in microorganisms that serves a nutritional role for obtaining nitrogen from urea, while doubling as a virulence factor for many pathogenic species of bacteria, including *Helicobacter pylori*, *Proteus mirabilis*, and *Klebsiella pneumoniae*[32–34] and fungi, such as *C. neoformans* and *Coccidioides posadasii*[26,35]. Conversion of urea to ammonia by these microbes contributes to pathogenesis by eliciting tissue damage and buffering low

pH environments[36]. Sufficient ammonia is produced by *H. pylori* that its presence in human breath is used for diagnosis of infection[37]. We detected a small but measurable amount of ammonia gas in the exhalant of mice bearing a heavy fungal burden of urease-positive *C. neoformans* cells in their lungs, establishing urease activity during cryptococcal infection. Ammonia has been found to act as a signaling molecule capable of altering growth and developmental outcomes for several fungal species[38–40]. Thus, when considering the actions of urease in virulence it is also important to consider the actions of its product, which can diffuse through tissues to act remotely during pathogenesis. We discovered an unanticipated role for urease beyond its role of modulating pH within phagolysosomes[5] as its volatile product, ammonia, increases pH at a distance to bolster melanization of neighboring cells. Alkaline pH affects melanin production through a multifaceted mechanism as it promotes the chemical reactions required to polymerize melanin precursors[10], and also increased the expression and cell wall localization laccase 1, the enzyme that catalyzes the reaction. When we compared the outcome of murine

**a**

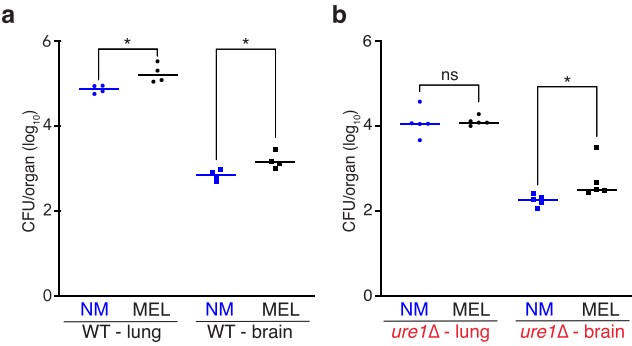

**b**

**Fig. 5 | Melanization promotes brain infiltration during murine infection.** Quantified fungal burden in the lungs and brains of mice 24 h after intravenous infection with $2 \times 10^5$ macrophages carrying NM or MEL WT ((**a**) $n = 4$, \*$p = 0.0167$ and 0.0300 for lung and brain, respectively; unpaired, two-sided parametric $t$-test) or $ure1\Delta$ ((**b**) $n = 5$, NS not significant, \*$p = 0.0414$ for brain; unpaired, two-sided parametric $t$-test). Source data are provided as a Source Data file.

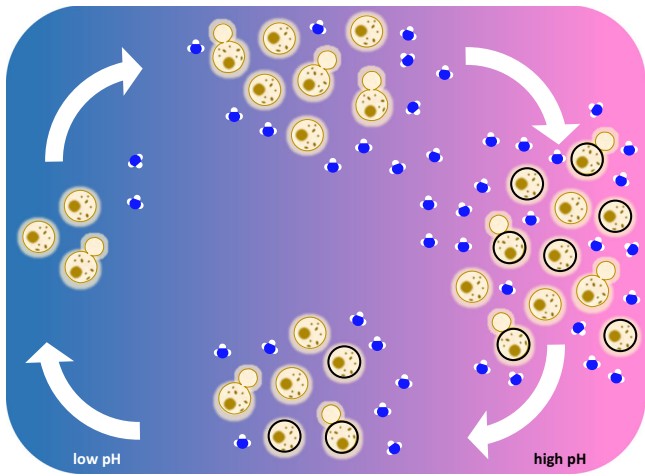

**Fig. 6 | Production of ammonia and melanin are inversely regulated through a feedback mechanism.** Replication of *C. neoformans* is favored at low pH but as population density increases, urease-mediated release of ammonia gas also increases to produce a more alkaline environment. The combination of enhanced melanization and slower replication at high pH prolongs persistence of cryptococcal cells within phagosomes to increase brain dissemination through a Trojan horse mechanism. Melanin deposition in the cell wall provides a feedback mechanism by impeding release of urease-carrying extracellular vesicles to decrease ammonia levels and restore optimal growth pH. Faster growth rates are expected to increase the incidence of lytic release from phagocytes to favor traversal of free yeast cells across the blood-brain barrier.

infection with macrophages carrying melanized and non-melanized *C. neoformans* cells, we found that melanization increased brain invasion, most likely by prolonging the survival of cryptococcal cells in macrophages to promote transmigration by a Trojan horse mechanism. Our findings reveal how a virulence mechanism operating in one cell can remotely modulate the virulence of neighboring cells, establishing a new mechanism for intracellular communication by microbial cells during infection.

Whereas urease promoted melanization, melanized cells exhibited reduced urease-mediated ammonia release. Growth of *C. neoformans* in the presence of melanin precursors does not affect urease expression[41] but melanin deposition on the cryptococcal cell wall reduces cell wall porosity[42] and permeability of cells to liposomes[15]. We found that melanization also impedes the release of extracellular vesicles from inside cells, and since urease is among the virulence factors transported in vesicles, this decreased urease activity for melanized cells. Since *C. neoformans* growth is favored at low pH[11] and inhibited under alkaline conditions (Fig. 2d), melanization reduces urease-containing vesicle release than in turn limits ammonia production to maintain pH within the optimal range for growth. We propose a model whereby the inverse relationship between urease and melanization provides a feedback mechanism that provides fungal cells with a means of rapidly fine-tuning urease activity in response to changing environmental conditions (Fig. 6).

The reciprocal interdependency between two virulence factors, urease activity and melanization, constitutes an emergent property in the pathogenesis of *C. neoformans* serving to maximize fungal survival during the temporal and spatial variations that occur in the course of infection. Following inhalation, invading cryptococcal cells are engulfed by pulmonary macrophages and this encounter has several possible outcomes. The host cell may kill the yeast or the fungal cells may survive and replicate within the phagocytic host cell leading to eventual lytic or non-lytic exocytosis[8]. Urease plays a central role in the outcome of this initial pathogen-host interaction as conversion of urea to ammonia increases phagosomal pH and decreases intracellular growth rates to favor persistence of cryptococcal cells within the host cell[5]. *C. neoformans* is unusual among pathogenic fungi in manifesting a remarkable neurotropism, such that most cases of cryptococcosis present as meningoencephalitis. Disseminated brain infection requires traversal of cryptococcal cells across the blood brain barrier either as free yeast cells or carried within macrophages by a 'Trojan horse' mechanism[43]. While urease plays a vital role in the former route[6,7], it may be less important for the latter route. Our results suggest that melanization, by virtue of prolonging survival in phagolysosomes, can

increase the frequency with which cells are carried into the brain inside macrophages to compensate for the loss of urease, at least partially.

In summary, we report that two major virulence factors of *C. neoformans* reciprocally modulate one another such that urease produces volatile ammonia that mediates effects on neighboring cells through action at a distance to increase pH and thereby promote melanization, which in turn reduces urease activity by inhibiting secretion of vesicles carrying the enzyme. Reciprocal modulation of melanin and urease production in turn yield fungal cells with different properties and resiliencies that can promote survival and dissemination during different phases of cryptococcosis. For microorganisms expressing multiple virulence factors, it is important to study these in the context of one another since their combined effects may not be predictable. Indeed, the intriguing possibility of additional interactions between individual virulence factors in *C. neoformans* warrants future investigation. For instance, urease may also be expected to impact capsule growth, since cells have larger capsules when grown at pH 7 compared to pH 5[44]. The interdependent relationship described here between urease and melanin production during cryptococcal infection manifests emergent properties that are neither predictable from, nor reducible to, their known individual functions. Coordination among individual virulence factors could contribute to virulence being an emergent property at the level of host-microbe interactions[45] and recognition of this complexity should guide future efforts in targeting microbial virulence.

## Methods
### Ethical statement
All animal procedures were performed with prior approval from the Animal Care and Use Committee (IACUC) of Johns Hopkins University, under approved protocol number MO21H124. Handling of mice and euthanasia with $CO_2$ in an appropriate chamber were conducted in accordance with guidelines on Euthanasia of the American Veterinary Medical Association. Johns Hopkins University is accredited by AAALAC International, in compliance with Animal Welfare Act regulations

and Public Health Service (PHS) Policy and has a PHS Approved Animal Welfare Assurance with the NIH Office of Laboratory Animal Welfare.

## Yeast strains
Wild-type *Cryptococcus neoformans* serotype A strain KN99α and the laccase 1 deletion mutant, *lac1Δ*, were obtained from the Fungal Genetics Stock Center gene deleted library[46]. The laccase 2 deletion mutant, *lac2Δ* (RPC 26)[13], was obtained from Joseph Heitman and J. Andrew Alspaugh (Duke University, NC). The urease-deficient mutant, *ure1Δ*[26], and the GFP-laccase 1 strain[14] were kindly provided by John Perfect (Duke University, NC) and Peter Williamson (NIH, MD), respectively.

## Ammonia diffusion assay
WT or *ure1Δ Cryptococcus neoformans* cells recovered from frozen 50% glycerol stocks by growth at 30 ˚C in yeast extract-peptone-dextrose (YPD) medium were sub-cultured into urea broth[9] at the cell density specified in each figure legend and transferred into the indicated well of a 24-well plate. After loading the remaining wells with cell-free urea broth, plates were sealed with parafilm and incubated at 30 ˚C for the indicated amount of time. Change in color of the cell-free media from yellow to pink was an indication of increased pH as a result of ammonia that diffused from cells growing in urea broth. Plates were photographed using a 12-megapixel camera and absorbance at 560 nm ($A_{560}$) was measured for each cell-free well using Softmax Pro 7.1 software with a SpectraMax iD5 multi-mode microplate reader (Molecular Dynamics). $A_{560}$ of a urea broth control was subtracted from each reading. A similar method was used to compare ammonia diffusion from non-melanized and melanized cells, except WT cells were pre-grown for 6 d in either minimal media (MM) comprised of 29.4 mM $K_2HPO_4$, 10 mM $MgSO_4$, 13 mM glycine, 15 mM D-glucose, and 3 μM thiamine at pH 5.5 or MM supplemented with 1 mM dopamine hydrochloride (DA, MilliporeSigma H8502) before being sub-cultured at a density of $5 \times 10^7$ cells/mL in urea broth.

## Measurement of ammonia produced by cultures
WT cells were grown in urea broth at 30 ˚C for 24 h with either the same starting cell density of $1 \times 10^8$ cells/mL and a range of urea concentrations or with a constant urea concentration of 2% and a range of starting cell densities. A *ure1Δ* control culture was inoculated at a cell density of $1 \times 10^8$ cells/mL in urea broth with 2% urea. The sensor of a BT-5800G ammonia gas detector with a range of 0–200 ppm was held directly above a 1.5 mL conical tube containing 1 mL of each culture for 30 s and readings were recorded. Measurements for each WT culture were corrected by subtracting the low background measurement of the *ure1Δ* control and then expressed as a percentage of the maximum of 200 ppm. $A_{560}$ of cell-free supernatants collected by centrifugation at 2500 *g* for 5 min were measured, subtracted for background absorbance of cell-free urea broth, and expressed as a percentage of the maximum reading.

## Stimulation of melanization by ammonia
MM supplemented with 0.1 or 1 mM DA and 1.5% agar was dispensed into the indicated wells of a 48-well plate and allowed to solidify. Each well of DA-supplemented agar was spotted with $5 \times 10^5$ phosphate buffered saline (PBS)-washed WT cells and then either WT or *ure1Δ* were cultured in urea broth at a cell density of $5 \times 10^7$ cells/mL in wells A1-A6 of the same plate such that melanization was assayed at a distance of 13, 26, 39, 52, 65, 78, or 91 mm. Parafilm-wrapped plates were incubated at 30 °C for 24 h and then photographed using a 12-megapixel camera. Images were converted to grayscale images using Adobe Photoshop and the intensity of pigmentation for each spot was quantified using Image Studio Lite 5.2 software (Li-Cor Biosciences). To derive the net increase in melanization imparted by the diffusion of ammonia, pigmentation measurements for each well in a *ure1Δ*-plate

were subtracted from that of the corresponding well in a WT-plate. A variation of this assay was used to compare melanization by WT, *lac1Δ*, and *lac2Δ* on 1 mM DA-supplemented agar adjacent to a well containing WT or *ure1Δ* in urea broth as described above.

## Melanization assays in 4-well plates
MM supplemented with 1 mM DA, 5 mM ascorbic acid (to minimize DA self-polymerization that occurs at high pH), and a universal buffer system consisting of 20 mM Tris-HCl, 20 mM Bis-Tris, and 20 mM sodium acetate, was adjusted to the desired pH and then mixed with 1.5% agar and dispensed into the wells of a 4-well plate. Solidified agar was spotted with $5 \times 10^5$ PBS-washed WT cells and after incubation at 30 °C for 48 h, plates were photographed, and pigmentation was quantified as described previously. A variation of this assay was used to compare melanization on DA-agar at pH 5.5 and 7.0 when one well contained urea broth supplemented with WT or *ure1Δ* cells or cell-free media. The pH of the agar in each well at the end of the experiment was measured using ColorpHast pH indicator strips.

## Growth curves near cells releasing ammonia
WT *C. neoformans* cells were sub-cultured in triplicate at a density of $1 \times 10^5$ cells/mL in 0.25X YPD broth in wells of a 48-well plate that were 13, 26, 39, or 52 mm from wells containing $5 \times 10^7$ cells in urea broth. The parafilm-sealed plate was incubated at 30 °C with continuous orbital shaking in a Spectromax iD5 plate reader and absorbance at 600 nm was read at 15 min intervals over a 36 h period.

## Expression and localization of GFP-laccase 1 near ammonia-producing cells
A *C. neoformans* strain expressing GFP-tagged laccase 1[14] was recovered from a frozen 50% glycerol stock by growth at 30 °C in yeast extract-peptone-dextrose (YPD) medium and then sub-cultured into MM supplemented with 10 μM $CuSO_4$ for 3 days at 30 °C. Cells were washed into fresh media and then dispensed into the wells of columns 2 through 6 of three 24-well plates to which $8 \times 10^7$ WT cells in urea broth were placed in each well of column 1 and then the plates were sealed with parafilm. Following incubation at 30 °C for 24 h, bright field and GFP fluorescence (excitation and emission wavelengths of 488 and 520 nm, respectively) images of live cells from columns 2 through 6 (distanced 19.3, 38.6, 57.9, 77.2, or 96.5 mm from cells producing ammonia) were captured using QCapture-Pro 6.0 software with a Retiga 1300 digital CCD camera (Teledyne Photometrics, Tucson, AZ) on an Olympus AX70 microscope (Olympus, Center Valley, PA) using an oil immersion 100X objective microscope. The cells from the 4 wells in each column were pooled and centrifuged for 5 min at 2500 *g* and then the pH of culture supernatants were measured using an accumet AB150 pH meter (Thermo Fisher Scientific, Waltham, MA). The cell pellets were resuspended in 500 μL media and fluorescence was quantified using excitation and emission wavelengths of 485 and 535 nm, respectively.

## Urease activity assay
WT *C. neoformans* cells grown at 30 ˚C for 4 d in either MM or MM supplemented with 1 mM DA were washed with PBS and then sub-cultured in triplicate into urea broth at a density of $1 \times 10^8$ cells/mL in tightly capped 14 mL tubes. After incubation at 30 ˚C for the indicated times, samples of culture were filtered through a Costar Spin-X 0.45 μm spin filter by centrifugation at 10,000 *g* for 2 min and $A_{560}$ of the cell-free filtrates were measured. The starting cell density of each culture was determined by plating a $10^{-6}$ dilution of each on Sabouraud dextrose (SAB) agar and counting colony forming units (CFU) after incubation at 30 °C for 2 d.

## Fluorescence-based lipid release assay
Extracellular vesicles in the supernatant of non-melanized and melanized *C. neoformans* cultures were quantified using the fluorescent

lipophilic probe, 1, 6-diphenyl-1, 3, 5-hexatriene (DPH), as described previously[47]. Five biological replicate cultures were inoculated at a cell density of $1 \times 10^8$ cells/mL in either MM or MM supplemented with 1 mM DA and incubated for 24 h at 30 °C before being sub-cultured into fresh media containing 10 μg/mL DPH. After incubation for an additional 24 h at 30 °C, cell-free supernatants were obtained by two sequential centrifugations at 2500 $g$ for 5 min. Fluorescence was quantified with excitation and emission wavelengths of 360 and 430 nm, respectively, and raw measurements were corrected by subtracting the background fluorescence measured for cell-free control media.

## Isolation of extracellular vesicles

Extracellular vesicles (EVs) were isolated from *C. neoformans* cells grown on solid media using a previously described protocol[16] with the following modifications. WT or *Δlac1* cells recovered from frozen 50% glycerol stocks by growth at 30 °C for 24 h in YPD medium were washed with PBS and then $1 \times 10^7$ cells were spread onto solid media plates prepared by dispensing 20 mL of a 1:1 mixture of 3% agar and either MM supplemented with 10 μM urea or MM supplemented with 10 μM urea and 1 mM DA. Following incubation at 30 °C for 48 h, cells were carefully recovered from the agar surface using a sterile inoculation loop. The cells from 2 plates were suspended in a total volume of 3 mL sterile PBS for each of 3 replicates for each variable. Cell number was quantified by counting dilutions of each sample using a hemocytometer. Following centrifugation at 4000 $g$ for 15 min at RT, supernatants were transferred to new tubes and centrifuged at 14,000 $g$ for 15 min at RT. After passage through a 0.8 μm filter, cell-free supernatants were centrifuged at 270,000 $g$ for 1 h at 10 °C. The EV pellets were resuspended in 100 μL 10 mM sodium phosphate, pH 7.0.

## Lipid quantification of EV preparations

Relative EV yields were compared by quantification of ergosterol content using an Amplex RED cholesterol assay kit (Invitrogen, A12216) following the provided protocol. Briefly, 25 μL of each EV preparation was allowed to react at 37 °C for 30 min and then fluorescence was quantified with excitation and emission wavelengths of 550 and 590 nm, respectively. After correcting raw measurements by subtracting the background fluorescence measured for a cholesterol-free control reaction, ergosterol concentration (μg/mL) was extrapolated from a cholesterol standard curve.

## Urease activity assay of EVs

Urease activity was quantified for 25 μL of each EV preparation by the Berthelot method using a urease activity assay kit (Sigma, MAK120) according to the manufacturer's instructions. Absorbance was read at 670 nm and urease activity (units/L) was calculated by extrapolation from an ammonia standard curve using the equation, ($A_{670}$ sample − $A_{670}$ blank) × n / (slope × t) where n is the dilution factor and, $t$ is time, and blank is the buffer-only assay control.

## *C. neoformans* intranasal mouse infections

Animal studies were performed using female C57BL/6 J mice (*Mus musculus*), aged 5–7 weeks (Jackson Laboratories) that were housed in a facility with a target room temperature of 22 °C and an acceptable range of 19–25 °C, a target room humidity of 42% and an acceptable range of 30–70%, and a 14.5 h light/9.5 h dark cycle. Mice were anesthetized by intraperitoneal injection of a mixture of 50 mg/kg ketamine and 5 mg/kg xylazine and then infected intranasally with ~ $5 \times 10^5$ WT or *ure1Δ* cells in a sterile 0.9% saline solution. CFU for each inoculum quantified by plating serial dilutions on SAB agar supplemented with 1% Penicillin-Streptomycin (P/S) were $5.8 \times 10^5$ and $6.3 \times 10^5$ for WT and *ure1Δ*, respectively. The sex of model animals was chosen for conformity to established protocols for *C. neoformans*

infections[5,26] and is not considered to influence the outcome of experiments.

## Ammonia measurement in mouse exhalant

The conical tip of a 50 mL centrifuge tube (Fisher Scientific, 06-443-19) was removed and the sensor of a BT-5800G ammonia gas detector was fitted tightly into the tube. While under ketamine/xylazine anesthesia (as described above), animals that had been infected with WT or *ure1Δ* for 20–22 d were placed inside the tube with the cap closed tightly and readings on the gas meter were recorded after 2 min.

## Mouse tissue and CFU measurements

Mice infected for 20–22 d with WT or *ure1Δ* were euthanized by intraperitoneal injection of a lethal dose of ketamine/xylazine (50 mg/kg for WT-infected animals that were already very ill from a high fungal burden in the lungs or 150 mg/kg for *ure1Δ*-infected animals). Infected lung tissue was exposed by dissection and probed with an MI-407 needle pH electrode (Microelectrodes Inc., NH) in combination with an accumet AB150 pH meter and Ag/AgCl reference electrode. Lungs were weighed using a DeltaRange analytical balance (Mettler Toledo AT261). Lung and brain tissue homogenized in sterile 0.9% saline by passage through a 100 μm cell strainer was serially diluted and plated on SAB + P/S agar to quantify CFU in each organ. Samples of homogenized lung tissue were also centrifuged at 10,000 $g$ for 5 min and pH of the resulting supernatant was measured using an accumet AB150 pH meter.

## Microscopy of mouse lung tissue sections

Lung tissue samples dissected from three mice infected with either WT or *ure1Δ* were fixed in neutral buffered 10% formalin at 4 °C and then submitted for histological analysis by the Johns Hopkins Oncology Tissue Services. Thin sections (4 μm) cut from paraffin embedded samples were mounted onto Superfrost Plus slides. Following dewaxing and rinsing with PBS, sections were stained with hematoxylin and eosin (H&E) and then scanned.

## Melanin ghosts isolated from lung tissue

Mouse lung tissue was homogenized in 3 mL 0.9% saline, mixed with an equal volume of concentrated (12 N) hydrochloric acid, and heated in a water bath at 95 °C for 1 h. Cooled samples were centrifuged at 10,000 $g$ for 30 min and acid resistant material was resuspended in 200 μL PBS. Wet mounts were prepared by applying 8 μL samples onto slides, examined under bright field illumination using an Olympus AX70 microscope with a 100X oil-immersion objective, and images were captured using QCapture-Pro 6.0 software with a Retiga 1300 digital CCD camera. From three slides each, the total number of melanin ghost particles identified were 29 and 8 for lung tissue derived from WT- and *ure1Δ*-infected mice, respectively.

## Murine infection by fungal-laden macrophages

Bone marrow derived macrophages (BMDMs) were extracted from C57BL/6 J mice and allowed to differentiate as described previously[5]. Cells were allowed to differentiate in BMDM media consisting of Dulbecco's Modified Eagle's Medium (DMEM), 10% Fetal Bovine Serum (FBS), 1% nonessential amino acids, 1% Penicillin-Streptomycin, 2 mM GlutaMAX, 1% HEPES buffer, 0.1% 2-Mercaptoethanol, and 20% L-929 cell-conditioned supernatant in 100 mm non-treated culture dishes (Corning, 430591) for 6–7 days at 37 °C in 9.5% $CO_2$. Differentiated cells were washed with Hank's Balanced Salt Solution (HBSS) and dislodged from culture dishes by treatment with Cellstripper for 10 min at 37 °C. Cells were washed into BMDM media and seeded at a density of $5 \times 10^5$ cells per well in 6-well plates and incubated at 37 °C in 9.5% $CO_2$ for 24 h. WT or *ure1Δ C. neoformans* cells grown for 3 days at 30 °C in either MM (non-melanized) or MM supplemented with 1 mM DA (melanized) were washed into BMDM media at a cell density of

$5 \times 10^5$ cells/mL and opsonized by incubation with 2 μg/mL 18B7 mAb[48] for 30 min at RT. The media in each well of BMDMs was replaced with media containing $1 \times 10^6$ *C. neoformans* cells and infection was allowed to proceed for 1 h at 37 °C. Infected BMDMs were washed 3 times with HBSS to remove free yeast cells and then washed into sterile 0.9% USP saline. Female C57BL/6 J mice, aged 5–7 weeks (Jackson Laboratories) were anesthetized by intraperitoneal injection of a mixture of 50 mg/kg ketamine and 5 mg/kg xylazine and infected with *C. neoformans*-laden BMDMs by retro-orbital injection of ~$2 \times 10^5$ cells. CFU for each inoculum were quantified by lysing BMDMs in sterile $H_2O$ and plating serial dilutions on SAB + P/S agar. After 24 h, infected animals were euthanized by $CO_2$ inhalation and dissected lungs and brains were homogenized in sterile PBS by passage through a 100 μm cell strainer. Serial dilutions of each tissue homogenate were plated on SAB + P/S agar and CFU counts were corrected for slight differences in CFU of inoculum.

## Statistical analysis

Data were graphed and analyzed for statistical significance using GraphPad Prism 9 software. Each data point in scatter plots denotes an independent biological replicate and horizontal lines denote the median. Statistical analyses were performed using unpaired, two-sided parametric t-tests for comparison of two data sets or an ordinary one-way analysis of variance (ANOVA) with Tukey's multiple comparisons test for analysis of more than two data sets (ns = not significant, $*p < 0.05$, $**p < 0.01$, $***p < 0.001$, $****p < 0.0001$).

## Reporting summary

Further information on research design is available in the Nature Portfolio Reporting Summary linked to this article.

# Data availability

The authors declare that the data supporting the findings of this study are available within the article. Source data are provided with this paper.

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

## Acknowledgements
We thank the Johns Hopkins Oncology Tissue Services for providing expert histological imaging. This work was supported by the National Institutes of Health, grant number R01-HL059842 (A.C.).

## Author contributions
R.P.B. and A.C. conceived the project. R.P.B. developed methodology, acquired and analyzed data and wrote the manuscript. A.C. edited the manuscript and acquired funding.

## Competing interests
The authors declare no competing interests.
