## [Peer Review File · Nature Communications]

Reciprocal modulation of ammonia and melanin production has implications for cryptococcal virulenceREVIEWER COMMENTS

Reviewer #1 (Remarks to the Author):

In this study, Baker and Casadevall describe the crosstalk between urease activity and melanization and how this interaction contributes to the pathogenicity of *Cryptococcus neoformans*. Through the use of several *in vitro* and *in vivo* approaches, the authors propose a novel reciprocal regulation mechanism between urease activity and consequent ammonia production and melanization, which in turn, impedes urease release from extracellular vesicles. Overall, this is an interesting and original study that provides new conceptual insights into how different fungal virulence factors collaborate to contribute to infection. Besides the identified mechanism, this study also suggests that therapeutic interventions that target the virulence composite of a pathogen instead of isolated virulence factors may be more effective. The results are provided in a clear and straightforward manner and the experimental approaches are sound and support, for the most part, the conclusions. I have a few comments that may help to improve the manuscript.

Major comments:

1. Melanin was identified as the target from urease activity, although it was not described how the authors immediately reached this conclusion. One important question that remains unanswered regards the possible effect of urease activity and ammonia release on the overall composition of the fungal capsule and cell wall and how this might also affect virulence.
2. Similar to the use of urease mutants to demonstrate the role of urease and ammonia, it would be important to demonstrate the involvement of melanin using genetic models. By resorting to albino strains of *C. neoformans*, the authors would be able to confirm the reciprocal link between melanization and urease activity, since an albino strain would in theory not respond to ammonia-derived cues.
3. There is no hard evidence showing that ammonia-induced melanization of yeast cells affects urease activity through decreased extracellular vesicle release. The authors base their conclusion on the use of a lipophilic probe, but it remains to be demonstrated that vesicle release is indeed impaired.
4. It also remains to demonstrate that melanization specifically affects the urease content of the vesicles and not any other cargo. Although in the end this might be difficult to prove, the authors should at least acknowledge this fact.
5. From a biological perspective, and particularly in a context of infection, I find it hard to conceive that the suggested mechanism would develop in a full loop, as it is not clear what would be the purpose for yeast cells to downregulate urease release and eventually their virulence potential. Additional discussion on the biological (and clinical) relevance of the identified mechanism would be helpful.

Minor comments:

6. Fig. 1a (right panel): it seems that the analysis indicates the correlation between the number of cells in the seeded wells and the 560nm absorbance of the unseeded wells, although this is not immediately clear from the text. I would suggest to reword it to make it clearer.
7. Fig. 2c. The text refers that "starting pH of dopamine-agar from 5.5 to 7 abrogated the ability of WT cells growing in urea broth to stimulate melanization of nearby cells". Should it not be the other way around? It seems that the higher pH actually promotes the melanization of adjacent cells.
8. Fig. 5. How many yeast cells from the initial 1×10^6 were engulfed by the macrophages and are thus delivered in the end to the mice?

Reviewer #2 (Remarks to the Author):

In the manuscript "reciprocal modulation of ammonia and melanin production has implications for cryptococcal virulence", the authors demonstrate that ammonia gas is produced as a consequence of the urease enzyme in *C. neoformans* both in in vitro culture systems and in mouse lungs. They go on to demonstrate that the production of ammonia gas promotes melanization with L-dopa as the substrate in vitro and demonstrate the presence of increased melanin in lung tissue infected with the wild type relative to the *ure1*Δ mutant. Melanization, in turn, reduces exosome mediated secretion of urease.

The data support a compelling narrative the activity of the virulence factor urease promotes the production of melanin pigment deposition on *C. neoformans*. An area in which there is a lack of narrative is in the molecular events triggered by ammonia gas that lead to increases in pigmentation.

Major critique:

1. Laccase, the enzyme that catalyzes the formation of melanin on the cell wall, is trafficked through the secretory pathway, and so there are many points of regulation at which ammonia could be operating. In work from the Williamson group (doi: 10.1128/IAI.01351-06), acidic pH was found to lead to an internal localization of laccase enzyme, whereas more physiological pH ranges promoted trafficking of laccase to the cell wall. Some assessment of the molecular localization or regulation of laccase production would increase the impact of the work.
2. *C. neoformans* encodes two laccase genes, LAC1 and LAC2. Under glucose starvation conditions, the primary catalyzer of melanin formation is LAC1. A clear role for LAC2 has not been elucidated. It would be important to know if the increase in melanin formation by ammonia gas is catalyzed by LAC1, the primary melanizer, or may represent a novel role for LAC2. Mutants of these genes are available.
3. In the BMDM experiment, the loaded macrophages are introduced into the mice and CFU taken 3 days later. How many of the cells isolated from brain at that timepoint are melanized? If the cells are replicating in the BMDMs or in the brain parenchyma, they may have different signals regulating melanization (glucose in the brain may be repressive). Here, using a *lac1* mutant may be informative – if the melanin deficient cells exhibit lower CFU in the presence and absence of *Ure1*, then you know the contribution of melanization apart from other effects of urease activity.

Reviewer #3 (Remarks to the Author):

The paper by Baker and Casadevall describes for the first time a reciprocal relationship between ammonia and melanin production during cryptococcal virulence. Overall, this is a new and exciting observation. The results are well described and illustrated and thoroughly discussed. However, as it stands, the paper does simply describe an interesting phenomenon. Few issues are below.

- 1) It is unclear whether the production of ammonia in the lung (~1.1 ppm) is similar to the amount of ammonia produced in cultured media (Fig. 1C).
- 2) Is this phenomenon specific to melanin? It will be interesting to provide results or comments on the relationship between ammonia production and other virulence factors (e.g. capsular production, phospholipase production, glucosylceramide synthesis...). This will show specificity of this important phenomenon.
- 3) What is the mechanism(s) by which ammonia increases melanin production? This can be exploited genetically or/and pharmacologically.
- 4) Similarly, what is the mechanism(s) by which melanin subsequently inhibits urease activity?

NCOMMS-22-33655-T: RESPONSES TO REVIEWERS' COMMENTS

Reviewer #1 (Remarks to the Author):

In this study, Baker and Casadevall describe the crosstalk between urease activity and melanization and how this interaction contributes to the pathogenicity of *Cryptococcus neoformans*. Through the use of several in vitro and in vivo approaches, the authors propose a novel reciprocal regulation mechanism between urease activity and consequent ammonia production and melanization, which in turn, impedes urease release from extracellular vesicles. Overall, this is an interesting and original study that provides new conceptual insights into how different fungal virulence factors collaborate to contribute to infection. Besides the identified mechanism, this study also suggests that therapeutic interventions that target the virulence composite of a pathogen instead of isolated virulence factors may be more effective. The results are provided in a clear and straightforward manner and the experimental approaches are sound and support, for the most part, the conclusions. I have a few comments that may help to improve the manuscript.

Major comments:

1. Melanin was identified as the target from urease activity, although it was not described how the authors immediately reached this conclusion. One important question that remains unanswered regards the possible effect of urease activity and ammonia release on the overall composition of the fungal capsule and cell wall and how this might also affect virulence.

Response: We thank the reviewer for these helpful comments and have addressed the concerns in the revised manuscript. To clarify our rationale in exploring the effect of urease-mediated ammonia release on melanization, we added the following statement to the Results section: "Since chemical oxidation of dopamine occurs more readily at alkaline pH¹⁰, we posited that increased pH due to ammonia production might promote melanin formation." We agree with the reviewer that ammonia also has the potential to impact virulence through modulation of the polysaccharide capsule but feel that exploring this possibility would require extensive additional investigation that is beyond the scope of our current study. We have provided the following comment in the revised Discussion section: "Indeed, the intriguing possibility of additional interactions between individual virulence factors in *C. neoformans* warrants future investigation. For instance, urease may be expected to impact capsule growth, since cells have larger capsules when grown at pH 7 compared to pH 5"

2. Similar to the use of urease mutants to demonstrate the role of urease and ammonia, it would be important to demonstrate the involvement of melanin using genetic models. By resorting to albino strains of *C. neoformans*, the authors would be able to confirm the reciprocal link between melanization and urease activity, since an albino strain would in theory not respond to ammonia-derived cues.

Response: We thank the reviewer for this insightful suggestion. The revised manuscript includes new data (Fig 2e) that compares the effect of nearby ammonia production on melanization by WT and two laccase deletion mutants of *C. neoformans*. The results demonstrate that deletion of

laccase 1 (*lac1Δ*), the primary laccase responsible for melanization, abrogates ammonia-mediated stimulation of melanization.

3. There is no hard evidence showing that ammonia-induced melanization of yeast cells affects urease activity through decreased extracellular vesicle release. The authors base their conclusion on the use of a lipophilic probe, but it remains to be demonstrated that vesicle release is indeed impaired.

Response: We are grateful to the reviewer for this suggestion and have taken the opportunity to perform a direct comparison of extracellular vesicle preparations from equivalent numbers of non-melanized and melanized cells. We used a recently developed and highly efficient protocol to isolate extracellular vesicles from WT and *lac1Δ* cells grown on either MM or MM supplemented with dopamine (Fig 3c) and found that the yield, as measured by ergosterol assay, was lower for melanized compared to non-melanized WT cells and urease activity was comparably decreased.

4. It also remains to demonstrate that melanization specifically affects the urease content of the vesicles and not any other cargo. Although in the end this might be difficult to prove, the authors should at least acknowledge this fact.

Response: We apologize for the misunderstanding as we did not intend to imply that melanization impedes the release of urease-containing vesicles specifically. Rather, we wished to convey that reduced vesicular transport from melanized compared to non-melanized cells accounts for the decrease in urease activity because urease is among the virulence factors released from vesicles. We have clarified this statement in the Results section as follows: “Thus, melanin in the cryptococcal cell wall impedes vesicular transport from cells and this results in lower urease activity because urease is one of the virulence factors secreted in extracellular vesicles ¹⁴.”

5. From a biological perspective, and particularly in a context of infection, I find it hard to conceive that the suggested mechanism would develop in a full loop, as it is not clear what would be the purpose for yeast cells to downregulate urease release and eventually their virulence potential. Additional discussion on the biological (and clinical) relevance of the identified mechanism would be helpful.

Response: We agree that it seems counterintuitive to limit urease since it promotes virulence, but it may be the case that unchecked urease activity could increase pH above optimal levels. We have added the following statement to the third paragraph of the Discussion: “Since *C. neoformans* growth is favored at low pH ¹¹ and inhibited under alkaline conditions (Fig. 2d), melanization reduces urease-containing vesicle release than in turn limits ammonia production to maintain pH within the optimal range for growth.” We also outline the clinical implications for our proposed feedback mechanism in the legend to Fig. 6, “The combination of enhanced melanization and slower replication at high pH prolongs persistence of cryptococcal cells within phagosomes to increase brain dissemination through a Trojan horse mechanism. Melanin deposition in the cell wall provides a feedback mechanism by impeding release of urease-carrying extracellular vesicles to decrease ammonia levels and restore optimal growth pH. Faster growth rates are expected to increase the incidence of lytic release from phagocytes to favor traversal of free yeast cells across the blood-brain barrier.”

Minor comments:

6. Fig. 1a (right panel): it seems that the analysis indicates the correlation between the number of cells in the seeded wells and the 560nm absorbance of the unseeded wells, although this is not immediately clear from the text. I would suggest to reword it to make it clearer.

Response: Thank you for the suggestion to clarify the text. The results section has been revised to read, "Measurement of absorbance at 560 nm (A_{560}) for each well revealed a direct linear relationship between the color change in the cell-free well and the number of cells in the adjacent well" and the y-axis of the graph in Fig 1a has been changed to Absorbance (560 nm) cell free well.

7. Fig. 2c. The text refers that "starting pH of dopamine-agar from 5.5 to 7 abrogated the ability of WT cells growing in urea broth to stimulate melanization of nearby cells". Should it not be the other way around? It seems that the higher pH actually promotes the melanization of adjacent cells.

Response: We apologize for the confusion. This statement was meant to convey that when the dopamine-agar is adjusted to pH 7 before starting the experiment, there is no further stimulation of melanization by the released ammonia. This section of the text has been revised to read as follows: "Whereas ammonia production by WT cells growing in urea broth stimulated melanization of nearby cells growing on pH 5.5 dopamine-agar, pre-adjustment of the media to pH 7 yielded comparable levels of pigmentation for WT, *ure1Δ* and cell-free control plates (Fig. 2c, left panel)."

8. Fig. 5. How many yeast cells from the initial 1×10^6 were engulfed by the macrophages and are thus delivered in the end to the mice?

Response: As described in the methods section, the macrophages were washed after infection to remove free *C. neoformans* cells and the resulting inoculum was $\sim 2 \times 10^5$ infected macrophages. This number has also been added to the Fig 5 legend for clarification.

Reviewer #2 (Remarks to the Author):

In the manuscript "reciprocal modulation of ammonia and melanin production has implications for cryptococcal virulence", the authors demonstrate that ammonia gas is produced as a consequence of the urease enzyme in *C. neoformans* both in in vitro culture systems and in mouse lungs. They go on to demonstrate that the production of ammonia gas promotes melanization with L-dopa as the substrate in vitro and demonstrate the presence of increased melanin deposits in lung tissue infected with the wild type relative to the *ure1Δ* mutant. Melanization, in turn, reduces exosome mediated secretion of urease.

The data support a compelling narrative the activity of the virulence factor urease promotes the production of melanin pigment deposition on *C. neoformans*. An area in which there is a lack of narrative is in the molecular events triggered by ammonia gas that lead to increases in pigmentation.

Major critique:

1. Laccase, the enzyme that catalyzes the formation of melanin on the cell wall, is trafficked through the secretory pathway, and so there are many points of regulation at which ammonia could be operating. In work from the Williamson group (doi: 10.1128/IAI.01351-06), acidic pH was

found to lead to an internal localization of laccase enzyme, whereas more physiological pH ranges promoted trafficking of laccase to the cell wall. Some assessment of the molecular localization or regulation of laccase production would increase the impact of the work.

Response: We thank the reviewer for recommending this important experiment. In our revised manuscript, we present an assessment of laccase expression and molecular localization using a *C. neoformans* strain expressing GFP-tagged laccase 1 (Fig 2f). With increasing proximity to ammonia-producing cells, higher alkalinity promoted cell wall localization of GFP-laccase 1 and increased GFP fluorescence that suggests upregulated protein expression.

2. *C. neoformans* encodes two laccase genes, LAC1 and LAC2. Under glucose starvation conditions, the primary catalyzer of melanin formation is LAC1. A clear role for LAC2 has not been elucidated. It would be important to know if the increase in melanin formation by ammonia gas is catalyzed by LAC1, the primary melanizer, or may represent a novel role for LAC2. Mutants of these genes are available.

Response: We are grateful to the reviewer for this insightful suggestion and have performed the recommended experiment comparing the effect of ammonia production on melanization by WT, *lac1Δ*, and *lac2Δ* strains. The results (Fig 2e) demonstrate that deletion of laccase 1, but not laccase 2, abrogates ammonia-mediated stimulation of melanization.

3. In the BMDM experiment, the loaded macrophages are introduced into the mice and CFU taken 3 days later. How many of the cells isolated from brain at that timepoint are melanized? If the cells are replicating in the BMDMs or in the brain parenchyma, they may have different signals regulating melanization (glucose in the brain may be repressive). Here, using a *lac1* mutant may be informative – if the melanin deficient cells exhibit lower CFU in the presence and absence of Ure1, then you know the contribution of melanization apart from other effects of urease activity.

Response: We shared the reviewer's concern regarding the potential loss of melanization and therefore limited the infection time in our BMDM experiment to only 24 hours (this is indicated in the methods section and in the legend for Fig 5). We note that *in vivo* melanization does not occur until day 2-3 (PMID: 11496012) so for this experiment, animals were infected with macrophages carrying cells that had been allowed to become fully melanized *in vitro*. In our revised manuscript, we edited the description of this experiment in the results section to read, "We measured significantly higher fungal burdens in the lungs and brains of animals 24 h post-infection with melanized compared to non-melanized cells (Fig. 5a)."

Reviewer #3 (Remarks to the Author):

The paper by Baker and Casadevall describes for the first time a reciprocal relationship between ammonia and melanin production during cryptococcal virulence. Overall, this is a new and exciting observation. The results are well described and illustrated and thoroughly discussed. However, as it stands, the paper does simply describe an interesting phenomenon. Few issues are below.

1) It is unclear whether the production of ammonia in the lung (~1.1 ppm) is similar to the amount of ammonia produced in cultured media (Fig. 1C).

Response: We thank the reviewer for drawing our attention to this need for clarification. We have added the following statement to the Results section: “By way of comparison, a comparable ammonia reading of ~1.5 ppm was measured for 2×10^6 *C. neoformans* cells growing in the presence of 2% urea (Fig 1c).”

2) Is this phenomenon specific to melanin? It will be interesting to provide results or comments on the relationship between ammonia production and other virulence factors (e.g. capsular production, phospholipase production, glucosylceramide synthesis...). This will show specificity of this important phenomenon.

Response: We share the reviewer’s curiosity regarding the effect ammonia may have on additional virulence factors in *C. neoformans* but addressing this possibility would require extensive additional investigation that is beyond the scope of our current study. This sentiment has been included in our revised Discussion section as follows: “Indeed, the intriguing possibility of additional interactions between individual virulence factors in *C. neoformans* warrants future investigation. For instance, urease may be expected to impact capsule growth, since cells have larger capsules when grown at pH 7 compared to pH 5”

3) What is the mechanism(s) by which ammonia increases melanin production? This can be exploited genetically or/and pharmacologically.

Response: We agree with the reviewer that it is important to elaborate on the mechanism by which ammonia increases melanin production. In our revised manuscript, we demonstrate that with increasing proximity to ammonia-producing cells, higher alkalinity promotes cell wall localization of GFP-laccase 1 and increased GFP fluorescence that suggests upregulated protein expression (Fig 2f). In addition to this effect on laccase expression and localization, the initial chemical reactions required to polymerize melanin precursors are also favored at high pH (doi:10.1021/acs.jpcc.8b02304) and would therefore be promoted by ammonia production. We have revised the Discussion section to explain that ammonia promotes melanization by a multifaceted mechanism.

4) Similarly, what is the mechanism(s) by which melanin subsequent inhibits urease activity?

Response: Since urease is transported and released from cells in extracellular vesicles, we had proposed that melanin reduces urease activity due decreased vesicular transport from melanized compared to non-melanized cells: “we found that the melanized cell wall similarly imparts a barrier to the release of extracellular vesicles as evidenced from the lower fluorescence of a lipophilic probe in culture supernatants of melanized compared to non-melanized cells (Fig. 3b).” In our revised manuscript, we extended this analysis by directly comparing extracellular vesicle preparations from equivalent numbers of non-melanized and melanized cells. We isolated extracellular vesicles from WT and *lac1*Δ cells grown on either MM or MM supplemented with dopamine (Fig 3c) and found that the yield, as measured by ergosterol assay, was lower for melanized compared to non-melanized WT cells and urease activity was comparably decreased.

REVIEWERS' COMMENTS

Reviewer #1 (Remarks to the Author):

The authors have done a substantial effort during the revision and have adequately addressed all my comments.

Reviewer #2 (Remarks to the Author):

The authors have satisfactorily addressed the concerns of this reviewer.

Reviewer #3 (Remarks to the Author):

The authors did a great job in revising the manuscript and they responded to my comments in a satisfactory manner

NCOMMS-22-33655B: RESPONSES TO REVIEWERS' COMMENTS

REVIEWERS' COMMENTS

Reviewer #1 (Remarks to the Author):

The authors have done a substantial effort during the revision and have adequately addressed all my comments.

Response: We appreciate the time and effort the reviewer has taken to reconsider our revised manuscript and we thank the reviewer for their favorable assessment.

Reviewer #2 (Remarks to the Author):

The authors have satisfactorily addressed the concerns of this reviewer.

Response: We thank the reviewer for reconsidering our revised manuscript and are pleased that we have adequately addressed the concerns of the reviewer.

Reviewer #3 (Remarks to the Author):

The authors did a great job in revising the manuscript and they responded to my comments in a satisfactory manner

Response: We are grateful to the reviewer for their careful consideration and positive response of our revised manuscript.